# Fluorescent and Colorimetric Dual-Mode Strategy Based on Rhodamine 6G Hydrazide for Qualitative and Quantitative Detection of Hg^2+^ in Seafoods

**DOI:** 10.3390/foods12051085

**Published:** 2023-03-03

**Authors:** Ziwen Zhang, Ran Han, Sixuan Chen, Feilin Zheng, Xinmiao Ma, Mingfei Pan, Shuo Wang

**Affiliations:** 1Key Laboratory of Food Quality and Health of Tianjin, Tianjin University of Science and Technology, Tianjin 300457, China; 2State Key Laboratory of Food Nutrition and Safety, Tianjin University of Science & Technology, Tianjin 300457, China

**Keywords:** Hg^2+^, R6GH, dual mode, fluorescence, visualization

## Abstract

In this study, a rapid fluorescent and colorimetric dual-mode detection strategy for Hg^2+^ in seafoods was developed based on the cyclic binding of the organic fluorescent dye rhodamine 6G hydrazide (R6GH) to Hg^2+^. The luminescence properties of the fluorescent R6GH probe in different systems were investigated in detail. Based on the UV and fluorescence spectra, it was determined that the R6GH has good fluorescence intensity in acetonitrile and good selective recognition of Hg^2+^. Under optimal conditions, the R6GH fluorescent probe showed a good linear response to Hg^2+^ (R^2^ = 0.9888) in the range of 0–5 μM with a low detection limit of 2.5 × 10^−2^ μM (S/N = 3). A paper-based sensing strategy based on fluorescence and colorimetric analysis was developed for the visualization and semiquantitative analysis of Hg^2+^ in seafoods. The LAB values of the paper-based sensor impregnated with the R6GH probe solution showed good linearity (R^2^ = 0.9875) with Hg^2+^ concentration in the range of 0–50 μM, which means that the sensing paper can be combined with smart devices to provide reliable and efficient Hg^2+^ detection.

## 1. Introduction

In recent years, the rapid development of industry has brought serious pollution to the natural and food production environment. Unlike other types of pollution, heavy metal pollution can circulate within the environment and have the characteristic of being nondegradable, thus causing more serious harm and impact on human beings [1]. Heavy metal ions are capable of denaturing proteins in the living organisms. When these harmful heavy metals accumulate and concentrate in the human body, they can produce an accumulation of toxicity and cause very serious diseases [2,3]. Heavy metal contamination in food has been an important factor affecting food safety. Mercury (Hg) is one of the common heavy metals that can easily cause environmental pollution [4]. Usually, microorganisms in the soil can methylate the Hg element, making it easy to easily be absorbed by microorganisms and thus enter the food chain. By accumulating in the human body for a long time, high concentrations of Hg can produce very serious hazards [5]. Therefore, it is necessary and significant to monitor and detect Hg levels in the environment and food samples.

Currently, analytical strategies such as mass spectrometry and spectroscopy based on large-scale instruments, such as inductively coupled plasma mass spectrometry (ICP-MS) [6,7,8], atomic absorption spectrometry (AAS) [9,10], and atomic fluorescence spectrometry (AFS) [11,12], are still the main means for accurate detection of heavy metals (including Hg) in various samples, and such methods have unparalleled advantages in terms of detection accuracy and sensitivity. However, such advanced instruments are usually expensive and large, and require relatively complex sample pretreatment processes, which are inadequate for low-cost screening, in situ detection, and large-scale penetration. To avoid these problems, there is an urgent need to establish a fast, sensitive, and portable detection method for heavy metal targets [13,14]. The paper-based sensing strategy based on fluorescent and colorimetric dual mode can achieve naked-eye visualization of the target and portable, low-cost semiquantitative detection [15,16]; on the other hand, this dual-mode spectral signal output can largely guarantee the reliability of the detection results.

Due to the long emission wavelength, low biotoxicity, and pronounced color change, rhodamine-based compounds are commonly used as fluorescent and colorimetric labeling reagents in visualization assays [17,18,19]. These compounds can specifically complex or bind metal ions or organic small molecules while processing certain recognition abilities. In particular, rhodamine derivatives with spiro ring structure have different optical properties in open and closed loops, making them ideal materials for the construction of optical sensors. It is worth noting that rhodamine 6G hydrazide (R6GH) is one of the most important intermediates of rhodamine compounds, which attracted extensive attention in heavy metal detection studies [20,21]. The amide spiral ring structure with rhodamine as the parent nucleus has an “On–Off” feature. When specific metal ions are added, the amide ring will be opened, resulting in the rupture of the organic dye and enhanced fluorescence [22]. In our previous work [23], the R6GH dye with a spiral ring structure was found to be opened by Pb^2+^, causing a significant fluorescence signal (Ex: 552 nm). Based on this, a dual-mode fluorescence and colorimetric detection strategy was further designed and constructed for the rapid and efficient detection of Pb^2+^ in water and food samples.

Since heavy metal Hg^2+^ can also trigger the fluorescence switch of the R6GH probe, this study continued to explore the fluorescence response performance of the R6GH probe to the heavy metal Hg^2+^ under different solution systems and thus developed an effective fluorescence analysis strategy for Hg^2+^ (Figure 1). The study further developed portable detection test strips that not only allowed for the naked-eye colorimetric semiquantitative analysis of the Hg^2+^ content but also can be combined with a portable color reader for the rapid screening of target Hg^2+^ in a large number of samples. This study is of great interest for the development of effective strategies for the on-site detection and large-scale screening of trace hazardous substances in food.

## 2. Materials and Methods

### 2.1. Reagents and Materials

The reagents and solvent nitrates of target Hg^2+^ and other ions Cu^2+^, V^2+^, Cd^2+^, Mn^2+^, Zn^2+^, Cr^3+^, Co^2+^, Ag^+^, and K^+^; hydrazine hydrate (85%); rhodamine 6G (R6G, 99.5%); and ethylenediaminetetraacetic acid (EDTA) were purchased from Shanghai Aladdin Biochemical Technology Co., Ltd. (Shanghai, China). Organic reagents such as tetrahydrofuran (THF), methanol (MeOH), ethanol (EtOH), and acetonitrile (ACN) were purchased from Sinopharm Chemical Reagent Company. All the reagents used for R6GH probe synthesis and analysis were of analytical grade or higher and were not further purified.

### 2.2. Instruments

A Shimadzu UV-vis spectrophotometer (Tokyo, Japan, UV-2600) was used to record UV absorbance data, and a Thermo fluorescence spectrometer (Boston, MA, USA, Lumina) was applied for fluorescence analysis. An electric blast dryer (WG II-45BE, Tianjin, China) and a Teflon digestion tank were used for sample pretreatment. An Agilent inductively coupled plasma mass spectrometer (ICP-MS) (Santa Clara, CA, USA, 7700x) was used to compare and validate the data from the Hg^2+^ analysis strategy established in this study. An ordinary quantitative filter paper was used in the visual analysis, and a portable colorimeter from FRU Weifu Optoelectronics (WR-10, Wuxi, China) was used for LAB analysis in the paper-based assays.

### 2.3. Preparation of R6GH Probe

The R6GH fluorescent probe was prepared according to the method reported [23] and is briefly described as follows: accurately weighed R6G (0.5 g) was fully dissolved in EtOH (30.0 mL) in a 100 mL round-bottom glass flask, and a solution of hydrazine hydrate (85%, 2.0 mL) was added for thorough mixing. The mixture was cooled to room temperature and magnetically stirred for 10 h until the color of the mixed solution disappeared. After filtering under reduced pressure and washing with EtOH three times, the white solid product was collected as the R6GH fluorescent probe.

### 2.4. Optimization of Dual-Mode Detection System

The prepared R6GH fluorescent probes were dispersed in THF/H_2_O (*v*/*v*, 1:1), ACN, and MeOH/H_2_O (*v*/*v*, 3:1), respectively. The concentration of R6GH was controlled at 1.0 mM as a stock solution and diluted to the desired concentration for testing. Subsequently, equal volumes of R6GH solutions at 20 μM concentration in the three systems were mixed with different concentrations (0–60 μM) of Hg^2+^ to obtain the resulting mixtures with Hg^2+^ concentrations ranging from 0 to 30 μM. The mixture was reacted at room temperature for 10 min and then analyzed by UV and fluorescence spectroscopy. By comparing the results, the optimal system for the detection of Hg^2+^ was finally determined and used in the subsequent experiments.

### 2.5. Detection Procedure for Hg^2+^

Different concentration gradients (0–10 μM) of Hg^2+^ were added to the ACN system of the R6GH probe (20 μM), thoroughly mixed, and left at room temperature for fluorescence intensity detection. The common metal ions Cu^2+^, V^2+^, Cd^2+^, Mn^2+^, Zn^2+^, Cr^3+^, Co^2+^, Ag^+^, and K^+^ (10 μM) in food were added to the R6GH probe solution in the ACN system in the same way, and after sufficient reaction, the fluorescence intensity was compared to assess the interference of different metal ions with Hg^2+^.

The reproducibility and reversible mechanism of the R6GH probe for the detection of Hg^2+^ were also investigated using ethylenediaminetetraacetic acid (EDTA) to assess the reversibility of R6GH for the detection of Hg^2+^.

### 2.6. Preparation of Test Strips for Visual Detection of Heavy Metal Ions

Test strips (1 cm × 1 cm) were infiltrated in the solution of the R6GH probe (20 µM) in the THF/H_2_O (*v*/*v*, 1:1) solution and left for 10 min at room temperature. The solvent was removed from the test strips and allowed to dry. The paper with the R6GH probe immobilized was cut into strips and fixed, infiltrated in Hg^2+^ standard solutions with different concentration gradients (0–100 µM), and naturally dried to obtain color-developed immobilized paper-based sensors that can be used for naked-eye or fluorescence (365 nm UV) analysis and comparison of Hg^2+^.

### 2.7. Actual Sample Pretreatment

Seafood samples of oysters, yellow croaker, and prawn were purchased from a local market in Tianjin and stored in a refrigerator at 4 °C. The samples were first nitrated by adding 0.5 g of the actual sample and 10.0 mL of HNO_3_ together at room temperature into a Teflon beaker for an overnight treatment, then boiled until all dissolved, and the cooled solution was centrifuged at 8000 r/min to obtain the supernatant, which was adjusted to pH 6.0 using the NaOH solution (1 mol/L) and diluted to 50 mL with ultrapure water to prepare different concentration gradients of Hg^2+^.

## 3. Results and Discussion

### 3.1. Optimization of Hg^2+^ Assay System

The prepared R6GH probe can react with the target Hg^2+^, which can open the ring structure it possesses, leading to the enhanced fluorescence. The fluorescence response of the R6GH probe to the target Hg^2+^ in three solutions of THF/H_2_O (*v*/*v*, 1:1), ACN, and MeOH/H_2_O (*v*/*v*, 3:1) was investigated, and the sensitivity and accuracy for the dual-mode detection of Hg^2+^ in different systems were compared based on the Hg^2+^-induced changes in color and fluorescence intensity of the R6GH probe observed under natural and UV light.

Figure 2 has shown the color change of the R6GH probe in the three detection systems under natural light and UV after Hg^2+^ induced the colorimetric and fluorescence switching of the R6GH probe in the “On” state. When Hg^2+^ was added into the solution of the R6GH probe, the solution color changed from colorless to pink under natural light, and the fluorescence in the solution changed from no fluorescence to bright yellow fluorescence under the excitation of 365 nm UV light. Meanwhile, with the increase in Hg^2+^ concentration (0–30 µM), the solution color of the R6GH probe gradually deepened and stabilized. From these results, the color change of R6GH in the THF/H_2_O (*v*/*v*, 1:1) system was more obvious, with higher chromogen and brighter fluorescence produced, which was easier to observe. Therefore, the THF/H_2_O (*v*/*v*, 1:1) solution was considered more suitable for the visualization and fluorescent colorimetric analysis of Hg^2+^ by the R6GH probe and used for the detection process of Hg^2+^ colorimetric test strips.

To further explore the Hg^2+^ recognition performance of the R6GH probe in fluorescence and colorimetric detection, the UV absorbance and fluorescence intensity of the R6GH probe were investigated in three systems under the colorimetric and fluorescence “On” states induced by Hg^2+^, and it was found that the UV absorption and fluorescence intensity significantly increased with the addition of Hg^2+^. As illustrated in Figure 3, the R6GH probe has no UV absorption and fluorescence emission ability but appeared a UV absorption band near 530 nm and a clear fluorescence emission peak near 556 nm. By comparing the UV absorption and fluorescence spectra in the three detection systems, the ACN system obtained the highest absorbance and fluorescence intensity. This was because the ACN was one nonprotonic solvent that did not provide nor spontaneously transfer protons in the reaction, thus having good solubility for metal cations. Meanwhile, considering the good solubility of ACN for the R6GH probe, it allowed Hg^2+^ to form ion-dipole bonds with the solvent system, thus increasing the contact area of the complexation reaction between the R6GH fluorescent probe and Hg^2+^, which made the reaction faster and more adequate.

### 3.2. Establishment of R6GH-Based Fluorescence Strategy for Hg^2+^ Detection

Hg^2+^ can ligand-complex with the O atom of -COOH and the N atom of -NH_2_ in the R6GH probe [24], which induces the conversion of the rhodamine-based amide spiro ring structure from “Off” to “On” state, resulting in the color change or fluorescence enhancement of the originally colorless and nonfluorescent R6GH probe. The fluorescence intensity of the R6GH probe in ACN gradually increased with the increase in Hg^2+^ concentration, having a good linear relationship with Hg^2+^ concentration in the range of 0–5 μM with R^2^ of 0.9888 (Figure 4A). The limit of detection (LOD, S/N = 3) of Hg^2+^ reached 2.5 × 10^−2^ μM, indicating that the prepared R6GH fluorescent probe can sensitively respond to Hg^2+^.

The selectivity of the R6GH fluorescent probe for Hg^2+^ in the ACN system was further evaluated for common metal ions (Cd^2+^, Mn^2+^, V^2+^, Cu^2+^, Zn^2+^, Cr^3+^, Co^2+^, Ag^+^, and K^+^) in the study. It was found that these metal ions did not significantly enhance the fluorescence intensity of R6GH. When Hg^2+^ (10 μM) induced the fluorescence conversion of the R6GH probe to “open loop”, its fluorescence intensity was as high as 9435.1, which indicated that Hg^2+^ had a relatively obvious fluorescence enhancement effect on the R6GH probe. Except for the selected Cd^2+^, Mn^2+^, V^2+^, and Cu^2+^, which had weak enhancement on the fluorescence of R6GH [23] (approximately 20–50% of the fluorescence intensity of the same concentration of Hg^2+^), the other tested ions could not produce fluorescence enhancement on the R6GH probe. Furthermore, the interference factor K (*F*_others_/*F*_Hg_^2+^) was used to compare and assess the interference of other metal ions on the fluorescence response of target Hg^2+^. Based on the fluorescence intensity of tested metal ions at 5.0 μM, the calculated K values 0.47 (Cd^2+^), 0.29 (Mn^2+^), 0.27 (V^2+^), 0.24 (Cu^2+^), 0.002 (Ag^+^), 0.0014 (Zn^2+^), 0.0012 (K^+^), 0.00094 (Co^2+^), and 0.0008 (Cr^3+^) were all less than 1.0, further demonstrating that the prepared R6GH probe has good selectivity for Hg^2+^, which provides a great feasibility and theoretical basis for its future practical application.

EDTA can coordinate with a variety of metal ions to form complexes and was used to examine the fluorescence reversibility of the R6GH probe [25]. The results showed that the fluorescence intensity of the R6GH–Hg^2+^ system to ACN sharply decreased after the addition of EDTA and even almost restored to the original state of the R6GH probe (Figure 4B). This indicated that EDTA could release Hg^2+^ from the R6GH–Hg^2+^ complex and turn off the fluorescence switch of the R6GH probe. When Hg^2+^ was added again, the fluorescence intensity of the solution recovered to be close to that of the R6GH probe solution when only Hg^2+^ was present, indicating that EDTA can cause the demetallization of the R6GH probe and regeneration of the spirolactam ring. After five cycles, the fluorescence of the R6GH probe solution did not significantly increase or decrease, indicating that the R6GH probe can release Hg^2+^ through competition with EDTA for reversible cycling of Hg^2+^ detection.

### 3.3. Detection of Hg^2+^ in Seafoods Using R6GH-Based Fluorescent Probes

To verify the application capability of the prepared R6GH-based fluorescent probe, seafoods including oysters, yellow croaker, and prawn were selected and spiked Hg^2+^ with different concentrations (0.5, 2.0, and 4.0 μM) to perform the recovery experiments. The spiked samples were simply pretreated and used for the constructed R6GH-based fluorescent probes and widely accepted ICP-MS methods for detection. As shown in Table 1, the proposed R6GH-based fluorescent probe obtained acceptable recoveries (88.0–108.3%) of Hg^2+^ in each selected seafood with RSDs all below 5% (*n* = 3). Compared with the results obtained from the conventional ICP-MS method, good correlation was achieved with r^2^ > 0.99 (Figure 5). These results indicated that the proposed R6GH probe-based fluorescence strategy was an ideal tool for providing accurate and reliable detection of Hg^2+^ in food matrices. Table 2 has compared the merits of different strategies for Hg^2+^ detection, signifying that the proposed R6GH-based probe can offer a rapid, sensitive, and effective strategy for Hg^2+^.

### 3.4. Development of Visualization Paper-Based Sensor for Hg^2+^ Detection

Fluorescence sensing analysis suffers from the shortcoming that single signal readings are susceptible to environmental and human factors; thus, there is an urgent need to develop new sensing platforms capable of rapid, on-site, and reliable heavy metal ion detection and analysis. Based on the constructed fluorescence sensing system of R6GH–Hg^2+^, an intelligent, low-cost, and portable paper-based sensing platform was developed to realize fluorescent and colorimetric dual-mode signal output for more accurate, reliable, and convenient detection of heavy metal Hg^2+^. As shown in Figure 6A, the paper-based sensor constructed with a filter paper infiltrated with the R6GH–Hg^2+^ solution as a substrate (the paper used in the experiment has blue background fluorescence) showed a colorless to light pink change (natural light) and a blue to yellow-green change (UV light, 365 nm) with increasing Hg^2+^ concentration from 0 to 100 μM.

For visualization of the results, colorimetric signals can be recorded and analyzed using a colorimeter to provide the LAB values of the test strips. The LAB color model consists of three elements, luminance L and the associated colors A and B [30,31]. The LAB color space defining the color change can be further linked to digital cameras and smartphones, thus facilitating remote monitoring and online analysis. Thus, the chromatic aberration parameter ∆*E* can be used as a reference for the visual detection of Hg^2+^ by paper-based sensors. The chromaticity difference at different concentrations of Hg^2+^ can be calculated according to the equation ($\Delta E=\sqrt{\Delta L^{2}+\Delta A^{2}+\Delta B^{2}}$) [32]. The results showed a good linear relationship (y = 0.439x + 6.472, R^2^ = 0.9875) between the chromatic aberration parameter ∆*E* and Hg^2+^ concentration for the paper-based sensor infiltrated using the R6GH–acetonitrile probe solution in the Hg^2+^ concentration range of 2.5–50 μM (Figure 6B). Therefore, the paper-based array sensor constructed based on fluorescence and colorimetric detection strategies combined with a smart device can achieve visualized semiquantitative detection of target Hg^2+^. This dual-mode sensing method based on R6GH enabled more convenient, reliable, and accurate analysis of target Hg^2+^.

## 4. Conclusions

In summary, this study successfully synthesized the R6GH probe that generates fluorescence and color signals with Hg^2+^ and developed a fluorescence-colorimetric dual-mode sensing platform that can be used for rapid, accurate, and sensitive detection of Hg^2+^ in seafood. This R6GH probe was used not only to construct a fluorescence analysis platform with good linearity, accuracy, and sensitivity for Hg^2+^ but also to develop a paper-based visual semiquantitative analysis strategy that can work in conjunction with a small and portable color reader, providing an ideal tool for rapid, low-cost, and convenient analysis of Hg^2+^. The R6GH fluorescent probe-based strategy and research method proposed in this study can be extended to the detection of other targets in other fields, providing a new direction for the research of high-performance and intelligent analytical strategies and detection devices.

## Figures and Tables

**Figure 1 foods-12-01085-f001:**
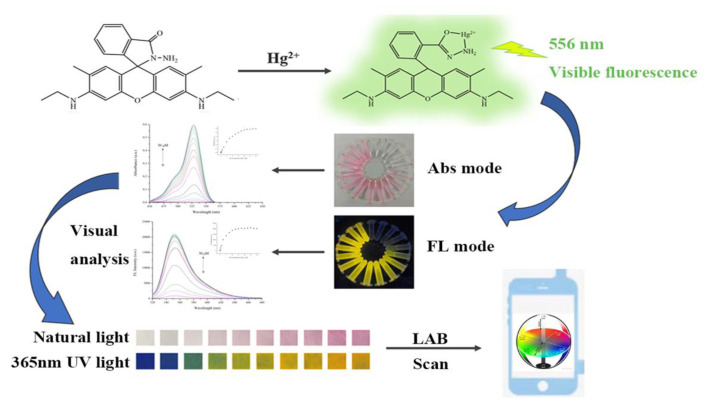
Schematic of R6GH synthesis and fluorescent and colorimetric dual-mode detection of Hg^2+^ for smartphone-integrated sensing system.

**Figure 2 foods-12-01085-f002:**
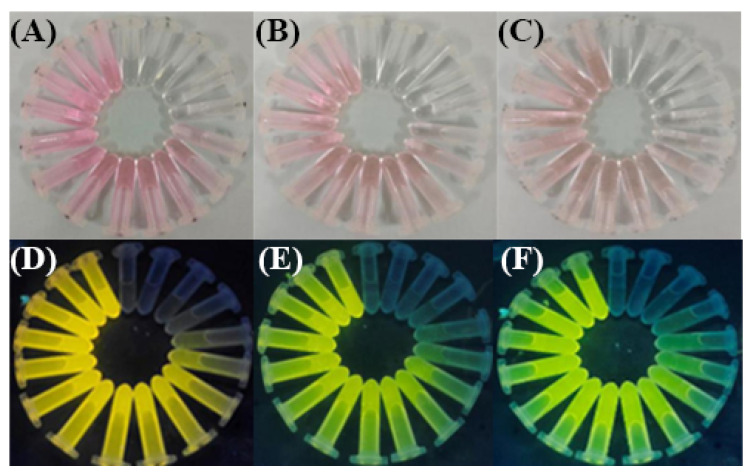
Color changes of Hg^2+^ induced R6GH probes in the three different solutions under natural light ((**A**): THF/H_2_O (*v*/*v*, 1:1); (**B**): ACN; (**C**): MeOH/H_2_O (*v*/*v*, 3:1)) and under 365 nm UV light ((**D**): THF/H_2_O (*v*/*v*, 1:1); (**E**): ACN; (**F**): MeOH/H_2_O (*v*/*v*, 3:1)).

**Figure 3 foods-12-01085-f003:**
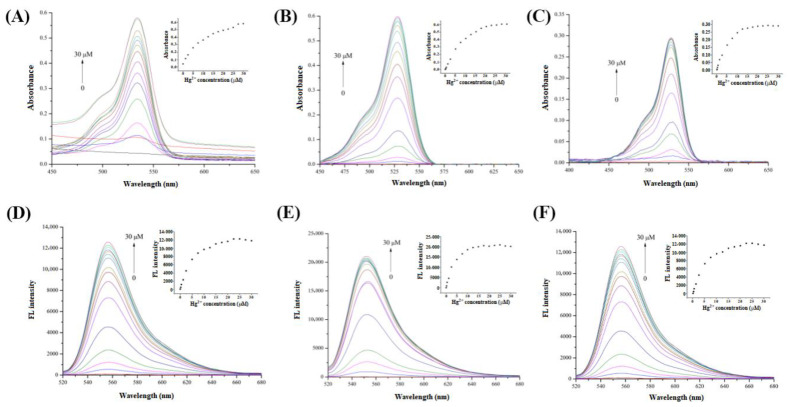
UV absorption spectra ((**A**) THF/H_2_O (*v*/*v*, 1:1); (**B**) ACN; (**C**) MeOH/H_2_O (*v*/*v*, 3:1)) and fluorescence spectra ((**D**) THF/H_2_O (*v*/*v*, 1:1); (**E**) ACN; (**F**) MeOH/H_2_O (*v*/*v*, 3:1)) of the R6GH probe (10 μM) in three tested solutions complexing with different concentrations of Hg^2+^ (0–30 μM). Inset: Relationship between R6GH absorbance or fluorescence intensity (Em: 556 nm) and Hg^2+^ concentrations.

**Figure 4 foods-12-01085-f004:**
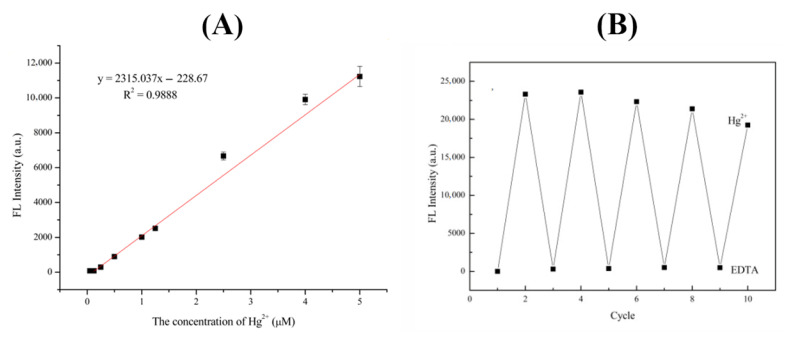
(**A**) Correlation between Hg^2+^ concentrations and R6GH fluorescence intensity. (**B**) Fluorescence reversibility of R6GH under alternate addition of Hg^2+^ and EDTA.

**Figure 5 foods-12-01085-f005:**
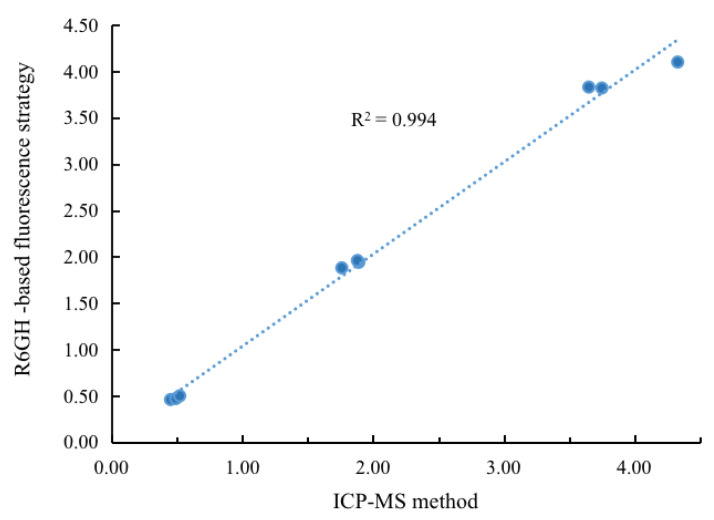
Correlation of Hg2+ detection results in real samples by R6GH-based fluorescence strategy and ICP-MS method.

**Figure 6 foods-12-01085-f006:**
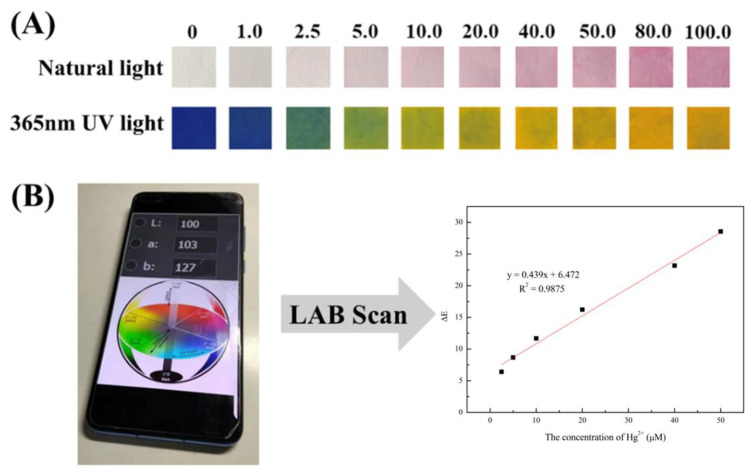
(**A**) Colors of immobilized paper-based sensors containing R6GH immersed in different concentrations of Hg^2+^ under natural and UV light. (**B**) Calibration curve of LAB ∆*E* value to Hg^2+^ concentration.

**Table 1 foods-12-01085-t001:** Results of Hg^2+^ detection in seafood samples using R6GH probe.

Seafood	Spiked Level (μM)	R6GH-Based Fluorescence Strategy			ICP-MS Method		
		Found (μM)	Recovery (%)	RSD (%, *n* = 3)	Found (μM)	Recovery (%)	RSD (%, *n* = 3)
Oysters	0.5	0.45	89.2	4.7	0.46	92.0	3.6
	2.0	1.76	88.0	3.5	1.88	94.0	3.2
	4.0	3.65	91.3	3.2	3.83	95.8	2.3
Yellow croaker	0.5	0.49	97.2	4.4	0.47	94.0	4.1
	2.0	1.89	94.7	2.8	1.94	97.0	3.1
	4.0	4.33	108.3	3.3	4.10	102.5	2.5
Prawn	0.5	0.52	103.6	4.0	0.50	100.0	3.7
	2.0	1.88	94.0	3.8	1.96	98.0	2.2
	4.0	3.75	93.8	2.4	3.82	95.5	1.9

**Table 2 foods-12-01085-t002:** Comparison of merits of different Hg^2+^ detection methods.

Methods	Materials	Linear Range	LOD	Required Time	Ref.
Multicapillary GC-ICP-MS	-	0.002–10 pg mL^−1^	0.08 pg	-	[6]
ICP-MS/MS	-	1.7–325.6 ng g^−1^	0.85 ng L^−1^	-	[7]
Fluorescent	Carbon nanodots	0–3 μM	4.2 nM	~10 min	[26]
Ultraviolet spectrophotometry	Gold Nanorods	285 nM–8.00 μM	112 nM	-	[27]
Ratiometric fluorescent paper	Dual-colored carbon dots	0–320 nM	0.14 nM	~3 min	[28]
Electrochemical biosensor	Poly-T oligonucleotides	1 nM–1.0 mM	100 pM	~30 min	[29]
Fluorimetry and visualization assay	R6GH	0–5 μM	0.025 μM	<10 min	This work

## Data Availability

Not applicable.

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
