# Peer review of "Fluorescent and Colorimetric Dual-Mode Strategy Based on Rhodamine 6G Hydrazide for Qualitative and Quantitative Detection of Hg2+ in Seafoods"

_foods, 2023, doi:10.3390/foods12051085_

Round 1
Reviewer 1 Report
In this paper, the authors provide additional metal detection of a R6GH dual-mode system that was already published by some of the same co-authors (10.1021/acs.jafc.1c02568). On the one hand, the emphasis of this this paper is on the suitability of the R6GH system on detecting Hg2+. On the other hand, their previous work finds the R6GH system to be selective at detecting Pb2+.
In general, a clear distinction between this manuscript and the previous paper is needed.
Can the authors explain the difference in values between Mn2+, Co2+, Cd2+, etc? The order in response in different in the current manuscript compared to the published work.
According to literature mercury is not present in ionic form (Hg2+) at pH 6, how can the authors measure Hg2+ at pH 6?
Can the authors describe how the fish and shrimp used in the previous work and the current work differs?
The Y-axis of the insets in Figure 2 are blurry and unreadable.
Author Response
Reviewer #1: In this paper, the authors provide additional metal detection of a R6GH dual-mode system that was already published by some of the same co-authors (10.1021/acs.jafc.1c02568). On the one hand, the emphasis of this this paper is on the suitability of the R6GH system on detecting Hg2+. On the other hand, their previous work finds the R6GH system to be selective at detecting Pb2+.
In general, a clear distinction between this manuscript and the previous paper is needed.
Reply:Thanks for your review. In the previous study, a R6GH-based dual-mode system was developed for the detection of Pb2+ (10.1021/acs.jafc.1c02568). Upon further investigation, Hg2+ were found to also induce a ring-opening reaction of R6GH, resulting in a fluorescent or visible light reaction. It is also of great interest to develop convenient test strips for the detection of Hg2+ using R6GH, which is the aim of this study.
- Can the authors explain the difference in values between Mn2+, Co2+, Cd2+, etc? The order in response in different in the current manuscript compared to the published work.
Reply:Thanks for your review. In the study, common metal ions including Cd2+, Mn2+, V2+, Cu2+, Zn2+, Cr3+, Co2+, Ag+, K+ were selected to evaluate the selectivity of the R6GH fluorescent probe. The obtained fluorescence intensity was used to calculate the interference factor K (Fothers/FHg2+). Figure 3B has illustrated the fluorescence response of R6GH to different metal ions, the order of which was obtained based on the high or low fluorescence response obtained. By comparing the results with those of previous studies, although the R6GH probe was prepared by the same method, the response to each metal ion was different from the previous measurements. To avoid causing ambiguity in the data, Figure 3B has been deleted and the manuscript has been revised as necessary. Thank you again for your review.
- According to literature mercury is not present in ionic form (Hg2+) at pH 6, how can the authors measure Hg2+ at pH 6?
Reply:Thanks for your review. We have checked the form of mercury ions present at different pH values and found no evidence that mercury ions could not be present at pH 6. In the study, Hg2+ solution was controlled to a pH of 6, and no abnormality was found.
- Can the authors describe how the fish and shrimp used in the previous work and the current work differs?
Reply:The experimental samples tested were selected based on the detection targets. Samples for the detection of Hg2+ and Pb2+ were selected from marine fish and sea shrimp because these samples are more susceptible to contamination and accumulation of toxic ions. The samples selected in both studies were purchased from local markets with different batches. Thank you for your comments.
- The Y-axis of the insets in Figure 2 are blurry and unreadable.
Reply:According to the comment, we have clarified the Figure 2 and ensured they are readable. Thanks for your review.
Reviewer 2 Report
In this study, a rapid fluorescent and colorimetric dual-mode detection strategy for Hg2+ in some fish was developed based on the cyclic binding of the organic fluorescent dye rhodamine 6G hydrazide (R6GH) to Hg2+.The work could be interesting but not supported at the validation level to give a strong scientific impact. Some considerations and revisions are needed:
Title and Abstract: Seafoods is not the correct word to indicate the sample analyzed. The category is fish. And when you talk about fish, for example in materials and methods, or table 1 specify which kind of fish.
In the literature there are already some examples of probes in this regard, specify the novelty compared to the present state of the art.
Lines 30, 35, 38, 66, 96, 185, 215, 264: Please, insert citation in square brackets
Lines 43, 63: Correct the citation into the brackets, please
Line 67: in the aim of the work specify for some fish
Line 76: Correct Figure 1 instead of Scheme 1, please
Line 83: Use formula or acronyms also for ethanol and acetonitrile
Lines 118-120: there are some repetitions in the sentence.
Figure 1. Specify the 3 different solutions in the caption and the different Hg2+ concentrations
Line 180: Delete one bracket
Figure 2: Specify the different Hg2+ concentrations in the caption
Line 194: “for” common
Paragraph 3.3. only recovery was assessed for validation? The data on this regard are scarce for scientific impact.
Author Contributions: please fill in according to the author guide.
Author Response
Reviewer 2
In this study, a rapid fluorescent and colorimetric dual-mode detection strategy for Hg2+ in some fish was developed based on the cyclic binding of the organic fluorescent dye rhodamine 6G hydrazide (R6GH) to Hg2+.The work could be interesting but not supported at the validation level to give a strong scientific impact. Some considerations and revisions are needed.
Reply:Based on the reviewers' comments, the manuscript was revised in detail and the changes were marked in color. Please refer to the revised manuscript for details. Thank you again for your review.
- Title and Abstract: Seafoods is not the correct word to indicate the sample analyzed. The category is fish. And when you talk about fish, for example in materials and methods, or table 1 specify which kind of fish.
Reply:Thanks for your review. We have checked the meaning of "seafood" and concluded that it refers to fish, shrimp and other foods from the sea. The samples selected for this study were oysters, yellow croaker and prawn, and therefore the use of "seafoods" was considered acceptable. In accordance with the comments, we added information on the type of fish and shrimp in the manuscript, hoping that it would meet the requirement.
- In the literature there are already some examples of probes in this regard, specify the novelty compared to the present state of the art.
Reply:Thanks for your review. This study prepared an R6GH fluorescent probe with stable response to Hg2+, and further developed a precise and sensitive fluorescence analysis strategy and a colorimetric detection test paper based on LAB analysis for Hg2+ in seafoods. These two strategies provide new options for the analysis and control of Hg2+ contaminants in food or environmental samples. Compared with other reported methods, the strategies developed in this study obtained higher accuracy and sensitivity, and also improved in reducing detection cost and improving throughput. In contrast to the application of other R6GH fluorescent probes, we prepared inexpensive colorimetric detection strips by immobilizing them on filter paper, which is advantageous for on-site, rapid and low-cost screening of contaminants. Thank you again for your review, the relevant changes have been marked in the manuscript, please refer to the revised manuscript.
- Lines 30, 35, 38, 66, 96, 185, 215, 264: Please, insert citation in square brackets
Lines 43, 63: Correct the citation into the brackets, please
Line 67: in the aim of the work specify for some fish
Line 76: Correct Figure 1 instead of Scheme 1, please
Line 83: Use formula or acronyms also for ethanol and acetonitrile
Lines 118-120: there are some repetitions in the sentence.
Figure 1. Specify the 3 different solutions in the caption and the different Hg2+ concentrations
Line 180: Delete one bracket.
Figure 2: Specify the different Hg2+ concentrations in the caption
Line 194: “for” common
Reply:According to the comments, the manuscript has been carefully revised and proofread in response to comments. Please see the revised manuscript for details. Thanks for your review.
- Paragraph 3.3. only recovery was assessed for validation? The data on this regard are scarce for scientific impact.
Reply:Thanks for your review. In the study, the spiked recoveries of the target substances in simulated real samples were measured, which is one of the most commonly used methods for evaluating food analysis strategies. The comparison of the measured results with the spiked amounts not only fully takes into account the influence of the food matrix, but also can reflect the ability of the established analytical strategy to be applied in real samples, i.e., the detection accuracy (recovery) and stability (RSD). We believe that the recovery and RSD data obtained in the study are representative of the merits of the developed analytical strategy. Necessary revisions were made in the revised manuscript. Please see the revised manuscript for details. Thank you again for your review.
- Author Contributions: please fill in according to the author guide.
Reply: According to the comments, the “Author Contributions” in the manuscript has been carefully revised. Thanks for your review.
Round 2
Reviewer 1 Report
1) I thank the reviewers for clarifying that, in both their previous (10.1021/acs.jafc.1c02568) and current works, the detection mechanism is due to a ring opening reaction. I urge the authors to clearly introduce their own previous work in the introduction. 10.1021/acs.jafc.1c02568 is reference [23] in the current work, but this reference is no mentioned as a very relevant work in the introduction. I believe a proper place would be in Line 64: “When specific metal ions are added, the amide ring will be opened, resulting in the rupture of the organic dye and enhanced fluorescence [22-23].” And, probably additional explanation of the system in [23] could be inserted after that sentence.
2) Author’s reply: Thanks for your review. We have checked the form of mercury ions present at different pH values and found no evidence that mercury ions could not be present at pH 6. In the study, Hg2+ solution was controlled to a pH of 6, and no abnormality was found.
New comment: authors are strongly recommended to include details of the preparation of Hg2+, the method for measuring Hg2+, and the evidence of Hg2+ concentration at different pH values. If possible, a curve of Hg2+ concentration vs. pH will be greatly appreciated. For instance, see Figure 3 in 10.1016/j.arabjc.2011.04.004 in which the concentration of Hg2+ concentration vs. pH is shown from theoretical calculations. Authors must describe the difference between the fact that they can measure Hg2+ at pH 6 in current work and Figure 3 in 10.1016/j.arabjc.2011.04.004.
3) New Figure 3 and axes in the insets are still unreadable. The authors can make it readable by increasing the font size of the axes in the subfigures and respective insets.
4) Also, the caption of the insets in the new Figure 3 needs to specify at which wavelength the fluorescence was obtained.
Author Response
- I thank the reviewers for clarifying that, in both their previous (10.1021/acs.jafc.1c02568) and current works, the detection mechanism is due to a ring opening reaction. I urge the authors to clearly introduce their own previous work in the introduction. 10.1021/acs.jafc.1c02568 is reference [23] in the current work, but this reference is no mentioned as a very relevant work in the introduction. I believe a proper place would be in Line 64: “When specific metal ions are added, the amide ring will be opened, resulting in the rupture of the organic dye and enhanced fluorescence [22-23].” And probably additional explanation of the system in [23] could be inserted after that sentence.
Reply:Thanks for the review. According to the comment, additional explanation on the R6GH probe in previous work was inserted in the introduction part. Please refer to the revised manuscript for details.
- Author’s reply: Thanks for your review. We have checked the form of mercury ions present at different pH values and found no evidence that mercury ions could not be present at pH 6. In the study, Hg2+ solution was controlled to a pH of 6, and no abnormality was found. New comment: authors are strongly recommended to include details of the preparation of Hg2+, the method for measuring Hg2+, and the evidence of Hg2+ concentration at different pH values. If possible, a curve of Hg2+ concentration vs. pH will be greatly appreciated. For instance, see Figure 3 in 10.1016/j.arabjc.2011.04.004 in which the concentration of Hg2+ concentration vs. pH is shown from theoretical calculations. Authors must describe the difference between the fact that they can measure Hg2+ at pH 6 in current work and Figure 3 in 10.1016/j.arabjc.2011.04.004.
Reply: Thanks for the comment. We are grateful for the literature provided. We have carefully referred to the provided literature (10.1016/j.arabjc.2011.04.004) and the Figure 3 in this report showed the species of Hg elements present in the aqueous system at different pH at Hg2+ concentration of 10-4 M. In our experiments, controlling the pH at 6.0 was performed on the supernatant after food nitrification. The supernatant was then fixed with water to 50 mL and was used to dilute the Hg2+ standard stock solution (1 mM, HNO3) to obtain Hg2+ working solutions (0.5, 2.0, 4.0 µM). This does not imply that the pH of Hg2+ working solution is 6.0. On request, the pH of the Hg2+ working solutions were determined using a pH meter to be 2.35 (4.0 µM), 2.79 (2.0 µM), 3.12 (0.5 µM), which is consistent with the literature (10.1016/j.arabjc.2011.04.004, Hg2+ + OH- ↔ Hg(OH)+ pK1 3.5; Hg(OH)+ + OH- ↔ Hg(OH)2 pK2 4.0). The Qc value ([Hg2+][OH-]2) was calculated as approximately 0.4~5*10-28, which is less than the Ksp(Hg(OH)2) value (3.0*10-26), which means that Hg2+ will not be precipitated in the study.
- New Figure 3 and axes in the insets are still unreadable. The authors can make it readable by increasing the font size of the axes in the subfigures and respective insets.
Reply:Thanks for the review. The font size of the axes in the subfigures and insets has been adjusted to be readable.
- Also, the caption of the insets in the new Figure 3 needs to specify at which wavelength the fluorescence was obtained.
Reply:Thanks for the review. The fluorescence intensity was obtained under 556 nm wavelength, which have been supplied in Figure 3 caption.
Reviewer 2 Report
11) In the literature there are already some examples of probes in this regard, report the state of art in the manuscript and specify the novelty in the main text, please.
22) Insufficient description of the validation procedure and results about it are presented in the manuscript.
33) Only spiked samples were analysed? Would be interesting to present results about real samples
Author Response
Comments and Suggestions for Authors
- In the literature there are already some examples of probes in this regard, report the state of art in the manuscript and specify the novelty in the main text, please.
Reply:Thanks for the review. Fluorescence analysis strategies based on rhodamine-based organic dyes have been widely studied for their excellent performance. In this study, based on the previous study, the fluorescence response of the prepared R6GH derivatives to Hg2+ in the closed-loop to open-loop state was examined, and then a fluorescence detection strategy based on the dual-signal response mode of the R6GH fluorescent probe for Hg2+ was developed, and semi-quantitative assay strips with LAB colorimetry were developed. Although there are other similar reports on rhodamine dye and detection of Hg2+, the dual signal output detection mode and low-cost colorimetric test strips proposed in this study are valuable. According to the comment, the manuscript has been revised and clarify the novelty of the study ( ).
- Insufficient description of the validation procedure and results about it are presented in the manuscript.
Reply:Thanks for the review. Comparison of results with standard accepted analytical methods is a common way to evaluate established analytical strategies. The results of the established R6GH-based fluorescence analysis strategy were verified by comparing with the results obtained by ICP-MS method with high confidence. Table 1 shows the test results of the two methods on the samples added with Hg2+, showing a good correlation. According to the comment, we have supplied more details for the comparison of these two strategies. A new figure and the recovery data of ICP-MS were supplied. Please refer to the revised manuscript.
3) Only spiked samples were analyzed. Would be interesting to present results about real samples.
Reply:Thanks for the review. We agree that the result of determining the target in real samples is more convincing. In this study, typical seafoods (oyster, yellow croaker, prawn) were selected for the spike-recovery test in order to demonstrate the application performance of the established R6GH probe-based detection strategy. Acceptable results were obtained at three spiked concentrations for three selected samples (recoveries: 88.0-108.3%, RSD less than 5%, n = 3), which were in good agreement with the results of the conventional ICP-MS method (r2 > 0.99). The selected food samples were detected to be free of Hg2+ by ICP-MS method before the addition of Hg2+. Even when Hg2+ was present in these samples, it was at very low levels, which necessitated the use of standard spiking methods for the relevant experiments.